# Clustering the Sketch: Dynamic Compression for Embedding Tables

**Henry Ling-Hei Tsang**[*]
Meta
henrylhtsang@meta.com

**Thomas Dybdahl Ahle**[*]
Meta[†]
Normal Computing
thomas@ahle.dk

## Abstract

Embedding tables are used by machine learning systems to work with categorical features. In modern Recommendation Systems, these tables can be very large, necessitating the development of new methods for fitting them in memory, even during training. We suggest Clustered Compositional Embeddings (CCE) which combines clustering-based compression like quantization to codebooks with dynamic methods like The Hashing Trick and Compositional Embeddings [Shi et al., 2020]. Experimentally CCE achieves the best of both worlds: The high compression rate of codebook-based quantization, but *dynamically* like hashing-based methods, so it can be used during training. Theoretically, we prove that CCE is guaranteed to converge to the optimal codebook and give a tight bound for the number of iterations required.

## 1 Introduction

Neural networks can efficiently handle various data types, including continuous, sparse, and sequential features. However, categorical features present a unique challenge as they require embedding a typically vast vocabulary into a smaller vector space for further calculations. Examples of these features include user IDs, post IDs on social networks, video IDs, and IP addresses commonly encountered in Recommendation Systems.

In some domains where embeddings are employed, such as Natural Language Processing [Mikolov et al., 2013], the vocabulary can be significantly reduced by considering "subwords" or "byte pair encodings". In Recommendation Systems like Matrix Factorization or DLRM (see Figure 2) it is typically not possible to factorize the vocabulary this way, leading to large embedding tables that demand hundreds of gigabytes of GPU memory [Naumov et al., 2019]. This necessitates splitting models across multiple GPUs, increasing cost and creating a communication bottleneck during both training and inference.

The traditional solution is to hash the IDs down to a manageable size using the Hashing Trick [Weinberger et al., 2009], accepting the possibility that unrelated IDs may share the same representation. Excessively aggressive hashing can impair the model's ability to distinguish its inputs, as it may mix up unrelated concepts, ultimately reducing model performance.

Another option for managing embedding tables is quantization. This typically involves reducing the precision to 4 or 8 bits or using multi-dimensional methods like Product Quantization and Residual Vector Quantization, which rely on clustering (e.g., K-means) to identify representative "code words" for each original ID. (See Gray and Neuhoff [1998] for a survey of quantization methods.) For instance, vectors representing "red", "orange", and "blue" may be stored as simply "dark orange" and "blue", with the first two concepts pointing to the same average embedding. See Section 1 for

---

[*]Equal contribution.
[†]Work done mostly at Probability at Meta.

37th Conference on Neural Information Processing Systems (NeurIPS 2023).

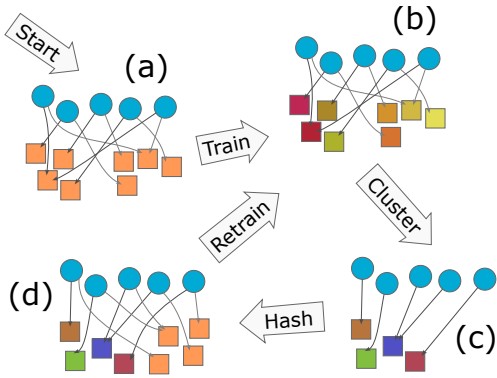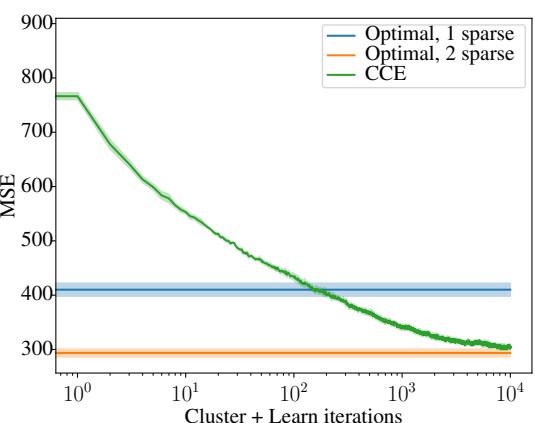

(a) **Single iteration of CCE**: **(a)** Starting from a random embedding table, each ID is hashed to a vector in each of 2 small tables. **(b)** During training, the embedding of an ID is taken to be the mean of the two referenced code words. **(c)** After training for an epoch, the vectors for all (or a sample of) the IDs are computed and clustered. This leaves a new small table in which similar IDs are represented by the same vector. **(d)** We can choose to combine the cluster centers with a new random table (and new hash function), after which the process can be repeated for an increasingly better understanding of which ID should be combined.

(b) **CCE for Least Squares**: The least squares problem is to find the matrix $T$ that minimizes $\|XT - Y\|_F$. To save space we may use codebook quantization, that is factorize $T \approx HM$, where $H$ is a sparse Boolean matrix, and $M$ is a small dense matrix. If we already know $T$ this is easy to do using the K-means algorithm, but if $T$ is too large to store in memory, we can find the compressed form directly using CCE, which we prove finds the optimal $H$ and $M$.

For the plot, we sampled $X \in \mathbb{R}^{10^4 \times 10^3}$ and $Y \in \mathbb{R}^{10^4 \times 10}$. We ran CCE and compared it with the optimal solution of first $T$ and then factorised it with either one or two 1s per row in $H$.

Figure 1: Clustered Compositional Embeddings is a simple algorithm which you run interspersed with your normal model training, such as every epoch of SGD. While the theoretical bound (and the least squares setting shown here) requires a lot of iterations for perfect convergence, in practice we get substantial gains from running 1-6 iterations.

an example. Clustering also plays a crucial role in the theoretical literature on vector compression [Indyk and Wagner, 2022]. However, a significant drawback of these quantization methods is that the model is only quantized *after* training, leaving memory utilization *during* training unaffected[3]

Recent authors have explored more advanced uses of hashing to address this challenge: Tito Svenstrup et al. [2017], Shi et al. [2020], Desai et al. [2022], Yin et al. [2021], Kang et al. [2021]. A common theme is to employ multiple hash functions, enabling features to have unique representations, while still mapping into a small shared parameter table. Although these methods outperform the traditional Hashing Trick in certain scenarios, they still enforce random sharing of parameters between unrelated concepts, introducing substantial noise into the subsequent machine learning model has to overcome.

Clearly, there is an essential difference between "post-training" compression methods like Product Quantization which *can utilize similarities between concepts* and "during training" techniques based on hashing, which are forced to randomly mix up concepts. This paper's key contribution is to bridge that gap: We present a novel compression approach we call "Clustered Compositional Embeddings" (or CCE for short) *that combines hashing and clustering while retaining the benefits of both methods.* By continuously interleaving clustering with training, we train recommendation models with performance matching post-training quantization, while using a fixed parameter count and computational cost throughout training, matching hashing-based methods.

In spirit, our effort can be likened to methods like RigL [Evci et al., 2020], which discovers the wiring of a sparse neural network during training rather than pruning a dense network post-training. Our work can also be seen as a form of "Online Product Quantization" Xu et al. [2018], though prior work

---

[3]Reducing the precision to 16-bit floats is often feasible during training, but this work aims for memory reductions much larger than that.

focused only on updating code words already assigned to the concept. Our goal is more ambitious: We want to learn *which* concepts to group together without ever knowing the "true" embedding for the concepts.

*Why is this hard?* Imagine you are training your model and at some point decide to use the same vector for IDs $i$ and $j$. For the remaining duration of the training, you can never distinguish the two IDs again, and thus any decision you make is permanent. The more you cluster, the smaller your table gets. But we are interested in keeping a constant number of parameters throughout training, while continuously improving the clustering.

In summary, our main contributions are:

- A new dynamic quantization algorithm (CCE) that combines clustering and sketching. Particularly well suited to optimizing the compression of embedding tables in Recommendation Systems.

- We use CCE to train the Deep Learning Recommendation System (DLRM) to baseline performance with less than 50% of the table parameters required by the previous state of the art, and a wobbling 11,000 times fewer parameters than the baseline models without compression.

- Using slightly more parameters, but still significantly less than the baseline, we can also improve the Binary Cross Entropy by 0.66%. Showing that CCE helps combat overfitting problems.

- We prove theoretically that a version of our method for the linear least-squares problem always succeeds in finding the optimal embedding table in a number of steps logarithmic in the approximation accuracy desired.

An implementation of our methods and related work is available at github.com/thomasahle/cce.

## 2 Background and Related Work

We show how most previous work on table compression can be seen in the theoretical framework of linear dimensionality reduction. This allows us to generalize many techniques and guide our intuition on how to choose the quality and number of hash functions in the system.

We omit standard common preprocessing tricks, such as weighting entities by frequency, using separate tables and precision for common vs uncommon elements, or completely pruning rare entities. We also don't cover the background of "post-training" quantization, but refer to the survey by Gray and Neuhoff [1998].

Theoretical work by Li et al. [2023] suggests "Learning to Count sketch", but these methods require a very large number of full training runs of the model. We only consider methods that are practical to scale to very large Recommendation Systems. See also Indyk and Wagner [2022] on metric compression.

### 2.1 Embedding Tables as Linear Maps

An embedding table is typically expressed as a tall skinny matrix $T \in \mathbb{R}^{d_1 \times d_2}$, where each ID $i \in [d_1]$ is mapped to the $i$-th row, $T[i]$. Alternatively, $i$ can be expressed as a one-hot row-vector $e_i \in \{0,1\}^{d_1}$ in which case $T[i] = e_i T \in \mathbb{R}^{d_2}$.

Most previous work in the area of table compression is based on the idea of sketching: We introduce a (typically sparse) matrix $H \in \{0,1\}^{d_1 \times k}$ and a dense matrix $M \in \mathbb{R}^{k \times d_2}$, where $k << d_1$, and take $T = HM$. In other words, to compute $T[i]$

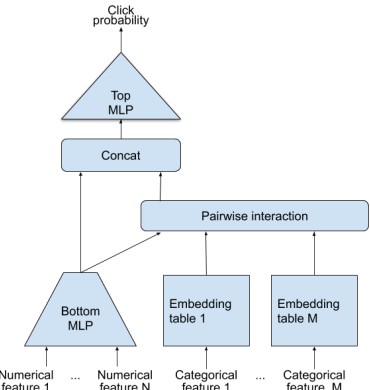

Figure 2: **Typical Recommendation System Architecture:** The DLRM model Naumov et al. [2019] embeds each categorical feature separately and combines the resulting vectors with pair-wise dot products. Other architectures use different interaction layers or a single embedding table for all categorical features, but the central role of the embedding table is universal. (Picture credit: Nvidia).

we compute $(e_i H)M$. Since $H$ and $e_i$ are both sparse, this requires very little memory and takes only constant time. The vector $e_i H \in \mathbb{R}^k$ is called "the sketch of $i$" and $M$ is the "compressed embedding table" that is trained with gradient descent.

In this framework, we can also express most other approaches to training-time table compression. Some previous work has focused on the "problem" of avoiding hash collisions, which intuitively makes sense as they make the model completely blind to differences in the colliding concepts. However, from our experiments, hashing does nearly as well as said proposed methods, suggesting that a different approach is needed. Sketching is a more general way to understand this.

**The Hashing Trick** [Weinberger et al., 2009] is normally described by a hash function $h : [d_1] \rightarrow [k]$, such that $i$ is given the vector $M[h(i)]$, where $M$ is a table with just $k \ll d_1$ rows. Alternatively, we can think of this trick as multiplying $e_i$ with a random matrix $H \in \{0, 1\}^{d_1 \times k}$ which has exactly one 1 in each row. Then the embedding of $i$ is $M[h(i)] = e_i H M$, where $HM \in \mathbb{R}^{d_1 \times d_2}$.

**Hash Embeddings** [Tito Svenstrup et al., 2017] map each ID $i \in V$ to the sum of a few table rows. For example, if $i$ is mapped to two rows, then its embedding vector is $v = M[h_1(i)] + M[h_2(i)]$. Using the notation of $H \in \{0, 1\}^{m \times n}$, one can check that this corresponds to each row having exactly two 1s. In the paper, the authors also consider weighted combinations, which simply means that the non-zero entries of $H$ can be some real numbers.

**Compositional Embeddings** (CE or "Quotient Remainder", Shi et al., 2020), define $h_1(i) = \lfloor i/p \rfloor$ and $h_2(i) = i \mod p$ for integer $p$, and then combines $T[h_1(i)]$ and $T[h_2(i)]$ in various ways. As mentioned by the authors, this choice is, however, not of great importance, and more general hash functions can also be used, which allows for more flexibility in the size and number of tables. Besides using *sums*, like Hash Embeddings, the authors propose element-wise *multiplication*[4] and *concatenation*. Concatenation $[T[h_1(i)], T[h_2(i)]]$ can again be described with a matrix $H \in \{0, 1\}^{d_1 \times k}$ where each row has exactly one 1 in the top half of $H$ and one in the bottom half of $H$, as well as a block diagonal matrix $M$. While this restricts the variations in embedding matrices $T$ that are allowed, we usually compensate by picking a larger $m$, so the difference in entropy is not much different from Hash Embeddings, and the practical results are very similar as well.

**ROBE embeddings** [Desai et al., 2022] are essentially Compositional Embeddings with concatenation as described above, but add some more flexibility in the indexing from the ability of pieces to "wrap around" in the embedding table. In our experiments, ROBE was nearly indistinguishable from CE with concatenation for large models, though it did give some measurable improvements for very small tables.

**Deep Hashing Embeddings** (DHE, Kang et al., 2021) picks 1024 hash functions $h_1, \ldots, h_{1024} : [d_1] \rightarrow [-1, 1]$ and feed the vector $(h_1(i), \ldots, h_{1024}(i))$ into a multi-layer perceptron. While the idea of using an MLP to save memory at the cost of larger compute is novel and departs from the sketching framework, the first hashing step of DHE is just sketching with a dense random matrix $H \in [-1, 1]^{d_1 \times 1024}$. While this is less efficient than a sparse matrix, it can still be applied efficiently to sparse inputs, $e_i$, and stored in small amounts of memory. Indeed in our experiments, for a fixed parameter budget, the fewer layers of the MLP, the better DHE performed. This indicates to us that the sketching part of DHE is still the most important part.

**Tensor Train** [Yin et al., 2021] doesn't use hashing, but like CE it splits the input in a deterministic way that can be generalized to a random hash function if so inclined. Instead of adding or concatenating chunks, Tensor Train multiplies them together as matrices, which makes it not strictly a linear operation. However, like DHE, the first step in reducing the input size is sketching.

**Learning to Collide** In recent parallel work, Ghaemmaghami et al. [2022] propose an alternate method for learning a clustering based on a low dimensional embedding table. This is like a horizontal sketch, rather than a vertical, which unfortunately means the potential parameter savings is substantially smaller.

Our method introduces a novel approach to dynamic compression by shifting from random sketching to *learned* sketching. This process can be represented as $e_i H M$, where $H$ is a sparse matrix and $M$ is a small dense matrix. The distinguishing factor is that we derive $H$ from the data, instead of relying on a random or fixed matrix. This adaptation, both theoretically and empirically, allows learning the same model using less memory.

---

[4]While combining vectors with element-wise multiplication is not a linear operation, from personal communication with the authors, it is unfortunately hard to train such embeddings in practice. Hence we focus on the two linear variants.

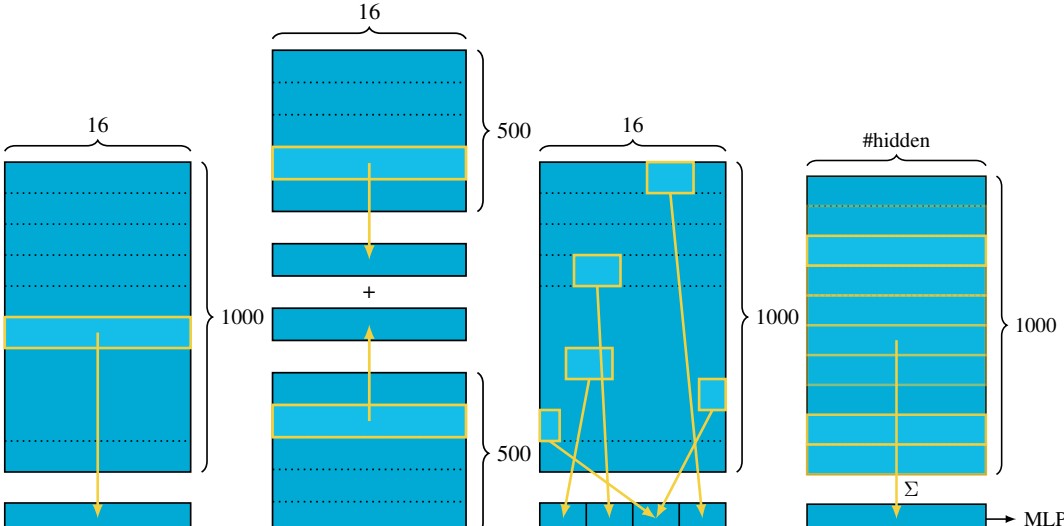

(a) **The Hashing Trick:** Also known as Count Sketch, each ID is hashed to one location in a table (here with 1000 rows) and it is assigned the embedding vector stored at the location. Many IDs are likely to share the same vector.

(b) **Hash Embeddings:** Each ID is hashed to two rows, one per table, and its embedding vector is assigned to be the sum of those two vectors. Here, we use two separate tables unlike in Tito Svenstrup et al. [2017].

(c) **ROBE:** Similar to CE with concatenation, but instead of using separate columns, it uses a continuous array from which the (in this case dim=4) pieces are picked. The pieces may even overlap.

(d) **Deep Hash Embeddings:** Computes a weighted sum of all the table entries, where the weights come from hashing the input 1000 times to $[-1, 1]$. The produced (hidden) vector is then sent through an MLP for further refinement.

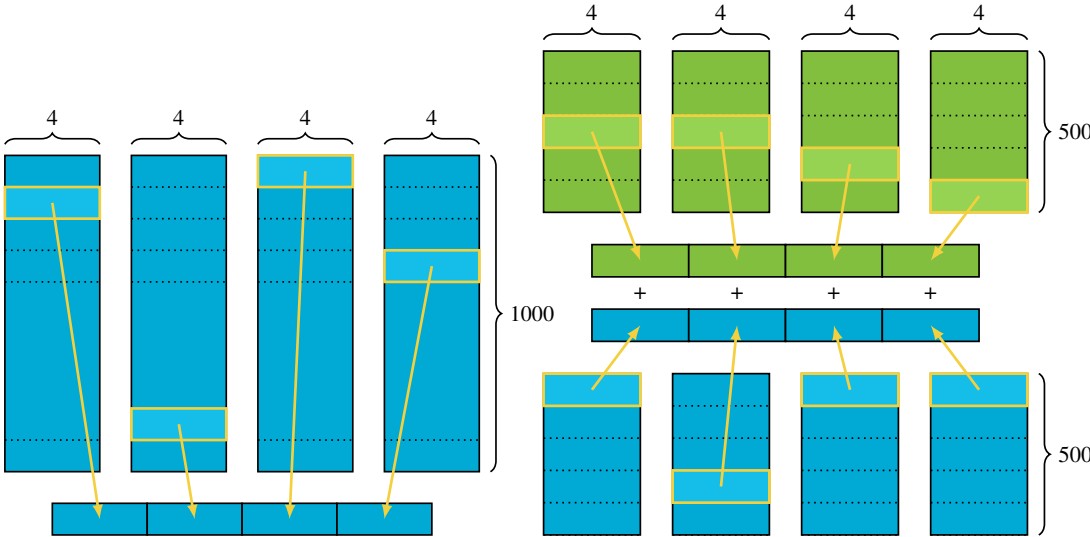

(e) **CE with concatenation:** In the hashing version of compositional embeddings (CE), each ID is hashed to a location in each of, say, 4 different tables. The four vectors stored there are concatenated into the final embedding vector. Given the large number of possible combinations (here $1000^4$), it is unlikely that two IDs get assigned the exact same embedding vector, even if they may share each part with some other IDs.

(f) **Clustered CE:** We can combine the sum hashing method of Tito Svenstrup et al. [2017] with the concatenation method of Shi et al. [2020]. Each ID then gets assigned a vector that is the concatenation of smaller sums. This is enhanced with the clustering idea shown in Section 1: In each of the four blocks, we apply clustering every epoch, setting the results in the green tables, and replacing the hash functions in the blue tables with new random values.

Figure 3: **The evolution of hashing-based methods for embedding tables.** The Hashing Trick and Hash Embeddings shown at the top, side by side with an equal amount of parameters. Next we introduce the idea of splitting the space into multiple concatenated subspaces. This is a classic idea from product quantization and reminiscent of multi-head attention in transformers. Finally in CCE we combine both methods in a way that allows iterative improvement using clustering.

# 3 Sparse and Dense CCE

The goal of this section is to give the background of Theorem 3.1, where we prove that CCE converges to the optimal assignments we would get from training the full embedding tables without hashing and clustering those.

We can't hope to prove this in a black box setting, where the recommendation model on top of the tables can be arbitrary, since there are pathological functions where only a full table will work. Instead, we pick a simple linear model, where the data is given by a matrix $X \in \mathbb{R}^{n \times d_1}$ and we want to find a matrix $T$ that minimizes the sum of squares, $\|XT - Y\|_F^2 = \sum_{i,j}((XT)_{i,j} - Y_{i,j})^2$.

We give two versions of the algorithm, a sparse method, which is what we build our experimental results on; and a dense method which doesn't use clustering, and replaces the Count Sketch with a dense normal distributed random matrix for which we prove optimal convergence. The dense algorithm itself is interesting since it constitutes a novel approximation algorithm for least squares regression with lower memory use.

| **Algorithm 1** Dense CCE for Least Squares | **Algorithm 2** Sparse CCE for Least Squares |
|---|---|
| **Require:** $X \in \mathbb{R}^{n \times d_1}, Y \in \mathbb{R}^{n \times d_2}, k \in \mathbb{N}$ | **Require:** $X \in \mathbb{R}^{n \times d_1}, Y \in \mathbb{R}^{n \times d_2}, k \in \mathbb{N}$ |
| 1: $H_0 = 0 \in \mathbb{R}^{d_1 \times 2k}$ | 1: $H_0 = 0 \in \mathbb{R}^{d_1 \times 2k}$ ▷ Initialize assignments |
| 2: $M_0 = 0 \in \mathbb{R}^{2k \times d_2}$ | 2: $M_0 = 0 \in \mathbb{R}^{2k \times d_2}$ ▷ Initialize codebook |
| 3: **for** $i = 0, 1, \dots$ **do** | 3: **for** $i = 0, 1, \dots$ **do** |
| 4: $\quad T_i = H_i M_i$ | 4: $\quad T_i = H_i M_i \in \mathbb{R}^{d_1 \times d_2}$ |
| 5: $\quad N \sim N(0,1)^{d_1 \times k}$ | 5: $\quad A = \text{kmeans}(T_i) \in \{0,1\}^{d_1 \times k}$ |
| 6: $\quad H_{i+1} = [T_i \mid N]$ | 6: $\quad C \sim \text{countsketch}() \in \{-1,0,1\}^{d_1 \times k}$ |
| 7: $\quad M_{i+1} = \arg\min_M \|XH_{i+1}M - Y\|_F^2$ | 7: $\quad H_{i+1} = [A \mid C] \in \mathbb{R}^{d_1 \times 2k}$ |
| 8: **end for** | 8: $\quad M_{i+1} = \arg\min_M \|XH_{i+1}M - Y\|_F^2$ |
| | 9: **end for** |

In the Appendix we show the following theorem guaranteeing the convergence of Algorithm 1:

**Theorem 3.1.** *Assume $d_1 > k > d_2$ and let $T^* \in \mathbb{R}^{d_1 \times d_2}$ be the matrix that minimizes $\|XT^* - Y\|_F^2$, then $T_i = H_i M_i$ from Algorithm 1 exponentially approaches the optimal loss in the sense*

$$E[\|XT_i - Y\|_F^2] \leq (1 - \rho)^{ik}\|XT^*\|_F^2 + \|XT^* - Y\|_F^2,$$

*where $\rho = \|X\|_{-2}^2/\|X\|_F^2 \approx 1/d_1$ is the smallest singular value of $X$ squared divided by the sum of singular values squared.*

We also show how to modify the algorithm to get an improved bound of $(1 - 1/d_1)^{ik}$ by conditioning the random part $H$ by the eigenspace of $X$. This means that after $i = O(\frac{d_1}{k}\log(1/\varepsilon))$ iterations we have a $1 + \varepsilon$ approximation to the optimal solution. Note that the standard least squares problem can be solved in $O(nd_1d_2)$ time, but one iteration of our algorithm only takes $O(nkd_2)$ time. Repeating it for $d_1/k$ iterations is thus no slower than the default algorithm for the general least squares problem, but uses less memory.

Some notes about Algorithm 2: In line 4 we compute $HM \in \mathbb{R}^{d_1 \times d_2}$ which may be very large. (After all the main goal of CCE is to avoid storing this full matrix in memory.) Luckily K-means in practice works fine on just a sample of the full dataset, which is what we do in the detailed implementation in Section 4.2.

In line 5, note that K-means normally returns both a set of cluster assignments and cluster centroids. For our algorithm we only need the assignments. We write those as the matrix $A$ which has $A_{id,j} = 1$ if $id$ is assigned to cluster $j$ and 0 otherwise. In the Appendix (Figure 5) we show how $A$ is a sparse approximation to $T$'s column space. Using the full, dense column space we'd recover the dense algorithm, Algorithm 1.

In line 6, $\text{countsketch}()$ is the distribution of $\{-1,0,1\}^{d_1 \times k}$ matrices with one non-zero per row. A matrix $C$ is sampled based on hash functions $h_i : [d_1] \to [k]$ and $s_i : [d_1] \to \{-1,1\}$ such that $C_{j,\ell} = s_i(j)$ if $h_i(j) = \ell$ and 0 otherwise. [5] We see that with $A$ and $C$ sparse, $XHM$ can be computed efficiently.

---

[5] We can ignore the sign, $\pm 1$ in most models if $M$ has mean 0. See Charikar et al. [2002].

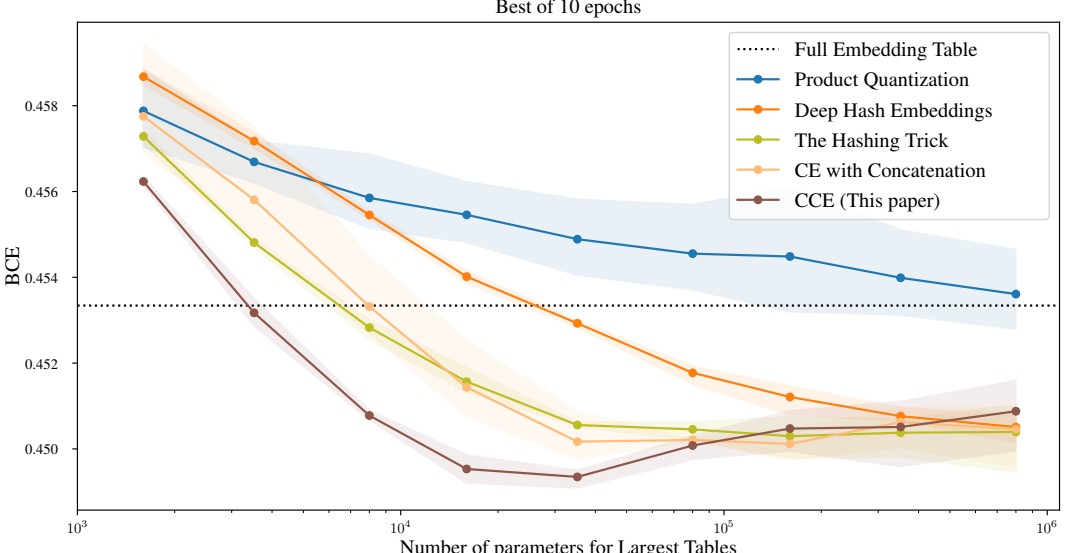

(a) **Best of 10 Epochs Test Loss on Criteo Kaggle**. We trained DLRM on the Criteo Kaggle dataset with different compression algorithms. Each of 27 categorical features was given its own embedding table, where we limited the number of parameters in the largest table as shown in the x-axis. Out of 10 epochs, we did early stopping at the minimum validation loss, plotting the test loss. We did independent repetitions with 3 seeds each, plotting the min, max and mean test losses. *Binary Cross-Entropy* was used as both the training and test loss. We also evaluated the AUC in Appendix G, but the results were indistinguishable. The *Full Embedding Table* used up to $16 \cdot 10^7$ parameters per table, as each categorical value is mapped to a unique embedding vector. This model over-fits immediately if trained for more than 1 epoch, which is why it does poorly compared to more parameter-restricted methods. For *Clustered Compositional Embeddings* we ran clustering once every epoch for the first 6 epochs, unless stopping earlier. *Product Quantization*, being a post-training quantization method, is never able to do better than the baseline model it is trained on. We tried fine-tuning the tables after PQ, but this approach immediately overfitted and gave terrible results. Interestingly this suggests CCE works as a regularization method for PQ.

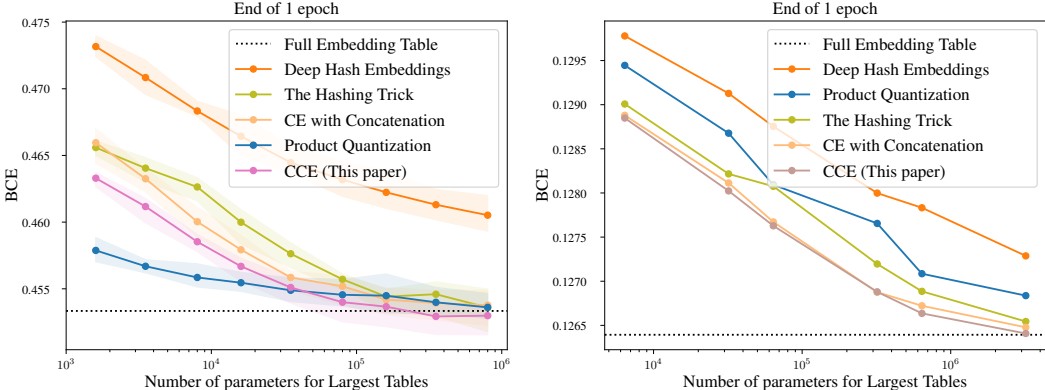

(b) **Kaggle dataset, 1 epoch**: Previous work, like Shi et al. [2020] only trained each method for one epoch, following DLRM, which did this to avoid over-fitting the baseline model. To compare with these results, we ran CCE clustering after $1/4$ and $1/2$ of an epoch. In this setting, the hashing-based methods were not able to compete with the baseline and product quantization, but CCE was able to do slightly better. Allowing a $300\times$ parameter reduction.

(c) **Terabyte dataset, 1 epoch**: We ran similar experiments on Criteo Terabyte dataset, which was expensive enough that we could only perform one repetition of each algorithm. Like on the Kaggle dataset, running for just 1 epoch is not enough to show an improvement in BCE over the baseline, but we are still able to save space. Curiously Product Quantization was basically no better than sketch-based methods for this dataset, which may be another reason CCE didn't perform great either.

Figure 4: CCE also improves models even when trained for just one epoch. This shows useful information is available from clustering even when only parts of the data have been seen.

Table 1: **Memory Reduction Rates Across all Datasets.** For each algorithm, dataset and epoch limit we measured the necessary number of parameters to reach baseline BCE. The compression ratios with ranges are estimated using degree 1 and 2 polynomial extrapolation. We can compare these results with reported compression rates from Desai et al. [2022] (ROBE) which gets $1000\times$ with multi epoch training; [Yin et al., 2021] (Tensor Train) which reports a $112\times$ reduction at 1 epoch on Kaggle; and Yin et al. [2021] which reports a $16\times$ reduction at 1 epoch with "Mixed Dimension methods" on Kaggle.

| Method | Dataset | Epochs | Embedding Compression |
|---|---|---|---|
| CCE (This Paper) | Criteo Kaggle | $\leq 10$ | $8{,}500\times$ |
| CE with Concatenation | Criteo Kaggle | $\leq 10$ | $3{,}800\times$ |
| The Hashing Trick | Criteo Kaggle | $\leq 10$ | $4{,}600\times$ |
| Deep Hash Embeddings | Criteo Kaggle | $\leq 10$ | $1{,}300\times$ |
| CCE (This Paper) | Criteo Kaggle | 1 | $212\times$ |
| CE with Concatenation | Criteo Kaggle | 1 | $127 - 155\times$ |
| The Hashing Trick | Criteo Kaggle | 1 | $78 - 122\times$ |
| Deep Hash Embeddings | Criteo Kaggle | 1 | $7 - 25\times$ |
| CCE (This Paper) | Criteo TB | 1 | $101\times$ |
| CE with Concatenation | Criteo TB | 1 | $25 - 48\times$ |
| The Hashing Trick | Criteo TB | 1 | $23 - 32\times$ |
| Deep Hash Embeddings | Criteo TB | 1 | $2 - 6\times$ |

## 4 Experiments and Implementation Details

Our primary experimental finding, illustrated in Table 1 and Figure 4a, indicates that CCE enables training a model with Binary Cross Entropy matching a full table baseline, using only a half the parameters required by the next best compression method. Moreover, when allocated optimal parameters and trained to convergence, CCE can yield a not insignificant $0.66\%$ lower BCE.

### 4.1 Experimental Setup

In our experiments, we adhered to the setup from the open-source Deep Learning Recommendation Model (DLRM) by Naumov et al. [2019], including the choice of optimizer (SGD) and learning rate. The model uses both dense and sparse features with an embedding table for each sparse feature. We modified only the embedding table portion of the DLRM code. We used two public click log datasets from Criteo: the Kaggle and Terabyte datasets. These datasets comprise 13 dense and 26 categorical features, with the Kaggle dataset consisting of around 45 million samples over 7 days, and the Terabyte dataset containing about 4 billion samples over 24 days.

We ran the Kaggle dataset experiments on a single A100 GPU. For the Terabyte dataset experiments, we ran them on two A100 GPUs using model parallelism. This was done mainly for memory reasons. With this setup, training for one epoch on the Kaggle dataset takes around 4 hours, while training for one epoch on the Terabyte dataset takes around 4 days. There was not a big difference in training time between the algorithms we tested. In total we used about 11,000 GPU hours for all the experiments across 5 algorithms, 3 seeds, 10 epochs and 9 different parameter counts.

### 4.2 CCE Implementation Details

In the previous section we gave pseudo code for "Sparse CCE for Least Squares" (Algorithm 2). In a general model, an embedding table can be seen as a data-structure with two procedures. Below we give pseudo-code for an embedding table with vocabulary $[d_1]$ and output dimension $d_2$, using $2kd_2$ parameters. The CCE algorithm is applied to each of $c$ columns, as mentioned in Figure 3f.

The value $c = 4$ was chosen to match Shi et al. [2020], but larger values are generally better, as long as the $h_i$ functions don't become too expensive to store. See Appendix E. The random hash functions $h_i'$ are very cheap to store using universal hashing. See Appendix D. The number of calls to Cluster was determined by grid search. See Appendix F for the effect of more or less clustering.

**Algorithm 3** Clustered Compositional Embeddings with $c$ columns and $2k$ rows

```
 1: class CCE:
 2:     method Initialize:
 3:         for i = 1 to c do:
 4:             h_i, h'_i ← i.i.d. ∼ random functions from [d_1] to [k]          ▷ H in Algorithm 2
 5:             M_i, M'_i ← i.i.d. ∼ N(0,1)^{k×d_2/c}
 6:
 7:     method GetEmbedding(id):
 8:         return CONCAT(M_i[h_i(id)] + M'_i[h'_i(id)] for i = 1 to c)
 9:
10:     method Cluster(items):
11:         for i = 1 to c do:
12:             T ← [M_i[h_i(id)] + M'_i[h'_i(id)] for id ∈ [d_1]]              ▷ See discussion below
13:             centroids, assignments ← K-MEANS(T)          ▷ Find k clusters and assign T to them
14:             h_i ← assignments
15:             M_i ← centroids
16:             h'_i ← random function from [d_1] to [k]
17:             M'_i ← 0^{k×d_2/c}
```

In line 12 we likely don't want to actually compute the embedding for every id in the vocabulary, but instead use mini batch K-Means with oracle access to the embedding table. In practice we follow the suggestion from FAISS K-means[Johnson et al., 2019] and just sample $256k$ ids from $[d_1]$ and run K-means only for this subset. The assignments from $id$ to nearest cluster center are easy to compute for the full vocabulary after running K-means. The exact number of samples per cluster didn't have a big impact on the final performance of CCE.

## 5 Conclusion

We have shown the feasibility of compressing embedding tables at training time using clustering. Our method, CCE, outperforms the state of the art on the largest available recommendation data sets. While there is still work to be done in expanding our theoretical understanding and testing the method in more situations, we believe this is an exciting new paradigm for dynamic sparsity in neural networks and recommender systems in particular.

Previous studies have presented diverging views regarding the feasibility of compressing embedding tables. Our belief is that Figure 4a, Figure 4b, and Figure 4c shed light on these discrepancies. At standard learning rates and with one epoch of training, it's challenging to make significant improvements over the DLRM baseline, corroborating the findings of Naumov et al. [2019]. However, upon training until convergence, it's possible to achieve parity with the baseline using a thousand times fewer parameters than typically employed, even with the straightforward application of the hashing trick. Nevertheless, in the realm of practical recommendation systems, training to convergence isn't a common practice. Our experiment, as illustrated in Figure 4a, proposes a potential reason: an excessive size of embedding tables may lead to overfitting for most methods. This revelation is startling, given the prevailing belief that "bigger is always better" and that ideal scenarios should allow each concept to have its private embedding vectors. We contend that these experimental outcomes highlight the necessity for further research into overfitting within DLRM-style models.

In this paper we analyzed only the plain versions of each algorithm. There are a number of practical and theoretical improvements one may add. All methods are naturally compatible with float16 and float8 reduced or mixed precision. The averaging of multiple embeddings may even help smooth out some errors. It is also natural to consider pruning the vocabularies. In particular in an offline setting we may remove very rare values, or give them a smaller weight in the clustering an averaging. However, in an online setting this kind of pruning is harder to do, and it is easier to rely on hash collisions and SGD to ignore the unimportant values. In our experiments we used 27 embedding tables, one for each categorical feature. A natural compression idea is to map all features to the same embedding table (after making sure values don't collide between features.) We didn't experiment with this, but it potentially could reduce the need for tuning embedding table sizes separately. A later paper by Coleman et al. [2023] report good results with this method. For things we tried, but didn't work, see Appendix A.

## Acknowledgment

We would like to thank Ben Ghaemmaghami from the University of Texas at Austin for fruitful discussions of learned hash functions. Thanks to Chunyin Siu for helpful discussions on the convergence of multi-step CCE. And finally to Michael Shi, Peter Tang, Summer Deng, Christina You, Erik Meijer, and the rest of the Probability group at Meta for helpful discussions of table embeddings and earlier versions of this manuscript.

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

## Reproducibility

The backbone recommendation model, DLRM by Naumov et al. [2019], has an open-source PyTorch implementation available on Github which includes an implementation of CE. For CCE you need a fast library for K-means. We recommend the open-sourced implementation by Johnson et al. [2019] for better performance, but you can also use the implementation in Scikit-learn [Pedregosa et al., 2011]. The baseline result should be straightforward to reproduce as we closely follow the instructions provided by Naumov et al. [2019]. For the CE methods, we only need to change two functions in the code: `create_emb` and `apply_emb`. We suggest using a class for each CE method; see Figure 3. For the random hash function, one could use a universal hash function or `numpy.random.randint`.

**Measuring the Embedding Compression factor**    The most important number from our experimental section is the "Embedding Compression" factor in Table 1. We measure this by training the model with different caps on of parameters in the embedding tables (See e.g. the x-axis in Figure 4a. E.g. if the Criteo Kaggle dataset has categorical features with vocabularies of sizes 10, 100 and $10^6$, we try e.g. to cap this at 8000, using a full embedding table for the small features, and a CCE table with $8000/16 = 500$ rows (since each row has 16 parameters). This corresponds to a compression rate of $(10 + 100 + 10^6)/(10 + 100 + 500) \approx 1639.5$. Or if we measure only the compression of the largest table, $10^6/500 = 2000$. Unfortunately there's a discrepancy in the article, which we only found after the main deadline, that uses the second measure in the introduction (hence the number 11,000x compression) where as Figure 4a uses the first measure (and thus the lower number 8,5000x).

For the experiments where we only train for 1 epoch, some methods never reach baseline BCE within the number of parameters we test. Hence the Compression Rates we report are based on extrapolations. For each algorithm we report a range, e.g. 127-155x for CE with Concatenation on Criteo Kaggle 1 epoch. Since the loss graphs tend to be convex, the upper bound (155x) is based on a linear interpolation (being optimistic about when the method will hit baseline BCE) and the lower bound (127x) is based on a quadratic interpolation, which only intersects the baseline at a higher parameter count.

**K-means**    For the K-means from FAISS, we use `max_points_per_centroid=256` and `niter=50`. The first parameter sub-samples the number of points to 256 times the number of cluster centroids ($k$), and is the recommended rate after which "no benefit is expected" according to the library maintainers. In practice we predict the right value will depend on the dimensionality of your data, so using the split into lower dimensional columns is beneficial. For niter we initially tried a larger value (300), but found it didn't improve the final test loss. We found on Kaggle for PQ, niter=50, BCE=0.455540 and niter=300, BCE=0.455537. For CCE (single epoch, single clustering at half an epoch), niter=50 gave BCE=0.45928 and niter=300 gave BCE=0.45905, so a very slight improvement, but not enough to make up for the extra training time.

**Datasets**    For our experiments, we sub-sampled an eighth of the Terabyte dataset and pre-hashed them using a simple modulus to match the categorical feature limit of the Kaggle dataset. For both Kaggle and Terabyte dataset, we partitioned the data from the final day into validation and test sets. Using the benchmarking setting of the DLRM code, the Kaggle dataset has around 300,000 batches while the Terabyte dataset has around 2,000,000 batches.

**Early stopping**    Early stopping is used when running the best of 10 epochs on the Kaggle dataset. We measure the performance of the model in BCE every 50,000 batches (around one-sixth of one epoch) using the validation set. If the minimum BCE of the previous epoch is less than the minimum BCE of the current epoch, we early stop.

**Deep Hash Embeddings**    We follow Kang et al. [2021] in using a fixed-width MLP with Mish activation. However, DHE is only described in one version in the paper: 5 layers of 1024 nodes per layer. For our experiments, we need to initialize DHE with different parameter budgets. We found that in general, DHE performs better with fewer layers when the number of parameters is fixed. However, we cannot use just a single layer, since that would be a linear embedding table, not an MLP. As a compromise, we fix the number of hidden layers to 2 and set the number of hashes to be the same as the dimension of the hidden layers.

For example, if we were allowed to use $64,000$ parameters with an embedding dimension of 64, then by solving a quadratic equation we get that the number of hashes and the dimension of the hidden layers are both 136. This gives us

$$\text{n\_hashes} \cdot \text{hidden\_dim} + 2 * \text{hidden\_dim}^2 + \text{hidden\_dim} \cdot \text{embedding\_dim} = 64192$$

parameters.

# A   What didn't work

Here are the ideas we tried but didn't work at the end.

**Using multiple helper tables**  It is a natural idea use more than one helper table. However, in our experiments, the effect of having more helper tables is not apparent.

**Circular clustering**  Based on the CE concat method, the circular clustering method would use information from other columns to do clustering. However, the resulting index pointer functions are too similar to each other, meaning that this method is essentially the hashing trick. We further discuss this issue in Appendix H.

**Continuous clustering**  We originally envisioned our methods in a tight loop between training and (re)clustering. It turned out that reducing the number of clusterings didn't impact performance, so we eventually reduced it all the way down to just one. In practical applications, with distribution shift over time, doing more clusterings may still be useful, as we discuss in Section 3.

**Changing the number of columns**  In general, increasing the number of columns leads to better results. However the marginal benefits quickly decrease, and as the number of hash functions grow, so does the training and inference time. We found that 4 columns / hash-functions was a good spot.

**Residual vector quantization**  The CCE method combines Product Quantization (PQ) with the CE concat method. We tried combining Residual vector quantization (RVQ) with the Hash Embeddings method from Tito Svenstrup et al. [2017]. This method does not perform significantly better than the Hash Embeddings method.

**Seeding with PQ**  We first train a full embedding table for one epoch, and then do Product Quantization (PQ) on the table to obtain the index pointer functions.

  We then use the index pointer functions instead of random hash functions in the CE concat method. This method turned out performing badly: The training loss quickly diverges from the test loss after training on just a few batches of data.

Here are some variations of the CCE method:

**Earlier clustering**  We currently have two versions of the CCE method: CCE half, where clustering happens at the middle of the first epoch, and CCE, where clustering happens at the end of the first epoch. We observe that when we cluster earlier, the result is slightly worse. Though in our case the CCE half method still outperforms the CE concat method.

**More parameters before clustering**  The CCE method allows using two number of parameters, one in Step 1 where we follow the CE hybrid method to get a sketch, and one in Step 3 where we follow the CE concat method. We thought that by using more parameters at the beginning, we would be able to get a better set of index pointer tables. However, the experiment suggested that the training is faster but the terminal performance is not significantly better.

**Smarter initialization after clustering**  In Algorithm 3 we initialize $M_i$ with the cluster centroids from K-means and the "helper table" $M_i' \leftarrow 0$. We could instead try to optimize $M_i'$ to match the residuals of $T$ as well as possible. This could reduce the discontinuity during training more than initializing to zeros. However, we didn't see a large effect in either training loss smoothness or the ultimate test score.

# B  Proof of the main theorem

$$
\begin{bmatrix}
0.417 & 0.720 \\
0.000 & 0.302 \\
0.147 & 0.092 \\
0.186 & 0.346 \\
0.397 & 0.539 \\
0.419 & 0.685 \\
0.204 & 0.878
\end{bmatrix}
\approx
\begin{bmatrix}
1 & 0 & 0 & 0 \\
0 & 1 & 0 & 0 \\
0 & 0 & 0 & 1 \\
0 & 1 & 0 & 0 \\
1 & 0 & 0 & 0 \\
1 & 0 & 0 & 0 \\
0 & 0 & 1 & 0
\end{bmatrix}
\begin{bmatrix}
0.411 & 0.648 \\
0.093 & 0.324 \\
0.204 & 0.878 \\
0.147 & 0.092
\end{bmatrix}
$$

Figure 5: **K-means as matrix factorization.** A central part of the analysis of CCE is the simple observation that K-means factors a matrix into a tall sparse matrix and a small dense one. In other words, it finds a sparse approximation the column space of the matrix.

Let's remind ourselves of the "Dense CCE algorithm" from Section 3: Given $X \in \mathbb{R}^{n \times d_1}$ and $Y \in \mathbb{R}^{n \times d_2}$, pick $k$ such that $n > d_1 > k > d_2$. We want to solve find a matrix $T^*$ of size $d_1 \times d_2$ such that $\|XT^* - Y\|_F$ is minimized – the classical Least Squares problem. However, we want to use memory less than the typical $nd_1^2$. We thus use this algorithm:

**Dense CCE Algorithm:**  Let $T_0 = 0 \in \mathbb{R}^{d_1 \times d_2}$. For $i = 1$ to $m$:

$$
\begin{aligned}
\text{Sample} \quad & G_i \sim N(0,1)^{d_1 \times (k-d_2)}; \\
\text{Compute} \quad & H_i = [T_{i-1} \mid G_i] \in \mathbb{R}^{d_1 \times k} \\
& M_i = \arg\inf_M \|XH_iM - Y\|_F^2 \in \mathbb{R}^{k \times d_2}. \\
& T_i = H_iM_i
\end{aligned}
$$

We will now argue that $T_m$ is a good approximation to $T^*$ in the sense that $\|XT_m - Y\|_F^2$ is not much bigger than $\|XT^* - Y\|_F^2$.

Let's consider a non-optimal choice of $M_i$ first. Suppose we set $M_i = \begin{bmatrix} I_{d_2} \\ M_i' \end{bmatrix}$ where $M_i'$ is chosen such that $\|H_iM_i - T^*\|_F$ is minimized. By direct multiplication, we have $H_iM_i = T_{i-1} + G_iM_i'$. Hence in this case minimizing $\|H_iM_i - T^*\|_F$ is equivalent to finding $M_i'$ at each step such that $\|G_iM_i' - (T^* - T_{i-1})\|_F$ is minimized.

In other words, we are trying to estimate $T^*$ with $\sum_i G_iM_i'$, where each $G_i$ is random and each $M_i'$ is greedily chosen at each step. This is similar to, for example, the approaches in Barron et al. [2008], though they use a concrete list of $G_i$'s. In their case, by the time we have $d_1/k$ such $G_i$'s, we are just multiplying $X$ with a $d_1 \times d_1$ random Gaussian matrix, which of course will have full rank, and so the concatenated $M$ matrix can basically ignore it. However, in our case we do a local, not global optimization over the $M_i$.

Recall the theorem:

**Theorem B.0.** *Given $X \in \mathbb{R}^{n \times d_1}$ and $Y \in \mathbb{R}^{n \times d_2}$. Let $T^* = \arg\min_{T \in \mathbb{R}^{d_1 \times d_2}} \|XT - Y\|_F^2$ be an optimal solution to the least squares problem. Then*

$$
\mathrm{E}\left[\|XT_i - Y\|_F^2\right] \le (1 - \rho)^{i(k-d_2)} \|XT^*\|_F^2 + \|XT^* - Y\|_F^2,
$$

*where $\rho = \|X\|_{-2}^2 / \|X\|_F^2$.*

Here we use the notation that $\|X\|_{-2}$ is the smallest singular value of $X$.

**Corollary B.1.** *In the setting of the theorem, if all singular values of $X$ are equal, then*

$$
\mathrm{E}\left[\|XT_i - Y\|_F^2\right] \le e^{-i\frac{k-d_2}{d_1}} \|XT^*\|_F^2 + \|XT^* - Y\|_F^2.
$$

*Proof of Theorem B.1.* Note that $\|X\|_F^2$ is the sum of the $d_1$ singular values squared: $\|X\|_F^2 = \sum_i \sigma_i^2$. Since all singular values are equal, say to $\sigma \in \mathbb{R}$, then $\|X\|_F^2 = d_1\sigma^2$. Similarly in this setting, $\|X\|_{-2}^2 = \sigma^2$ so $\rho = 1/d_1$. Using the inequality $1 - 1/d_1 \le e^{-1/d_1}$ gives the corollary. $\square$

*Proof of Theorem B.0.* First split $Y$ into the part that's in the column space of $X$ and the part that's not, $Z$. We have $Y = XT^* + Z$, where $T^* = \arg\min_T \|XT - Y\|_F$ is the solution to the least

squares problem. By Pythagoras theorem we then have

$$\mathrm{E}\left[\|XT_i - Y\|_F^2\right] = \mathrm{E}\left[\|XT_i - (XT^* + Z)\|_F^2\right] = \mathrm{E}\left[\|X(T_i - T^*)\|_F^2\right] + \|Z\|_F^2,$$

so it suffices to show

$$\mathrm{E}\left[\|X(T_i - T^*)\|_F^2\right] \leq (1 - \rho)^{i(k-d_2)}\|XT^*\|_F^2.$$

We will prove the theorem by induction over $i$. In the case $i = 0$ we have $T_i = 0$, so $\mathrm{E}[\|X(T_0 - T^*)\|_F^2] = \mathrm{E}[\|XT^*\|_F^2]$ trivially. For $i \geq 1$ we insert $T_i = H_i M_i$ and optimize over $M_i'$:

$$\begin{aligned}
\mathrm{E}[\|X(T_i - T^*)\|_F^2] &= \mathrm{E}[\|X(H_i M_i - T^*)\|_F^2] \\
&\leq \mathrm{E}[\|X(H_i[I \mid M_i'] - T^*)\|_F^2] \\
&= \mathrm{E}[\|X((T_{i-1} + G_i M_i') - T^*)\|_F^2] \\
&= \mathrm{E}[\|X(G_i M_i' - (T^* - T_{i-1}))\|_F^2]. \\
&= \mathrm{E}[\mathrm{E}[\|X(G_i M_i' - (T^* - T_{i-1}))\|_F^2 \mid T_{i-1}]] \\
&\leq (1 - \rho)^{k-d_2} \mathrm{E}[\|X(T^* - T_{i-1})\|_F^2] \\
&\leq (1 - \rho)^{i(k-d_2)}\|XT^*\|_F^2,
\end{aligned}$$

where the last step followed by induction. The critical step here was bounding

$$\mathrm{E}_G[\inf_M \|X(GM - T)\|_F^2] \leq (1 - \rho)^{k-d_2}\|XT\|_F^2,$$

for a fixed $T$. We will do this in a series of lemmas below. $\qquad\square$

We show the lemma first in the "vector case", corresponding to $k = 2, d_2 = 1$. The general matrix case follow below, and is mostly a case of induction on the vector case.

**Lemma B.2.** *Let $X \in \mathbb{R}^{n \times d}$ be a matrix with singular values $\sigma_1 \geq \cdots \geq \sigma_d \geq 0$. Define $\rho = \sigma_d^2 / \sum_i \sigma_i^2$, then for any $t \in \mathbb{R}^d$,*

$$\mathrm{E}_{g \sim N(0,1)^d}\left[\inf_{m \in \mathbb{R}} \|X(gm - t)\|_2^2\right] \leq (1 - \rho)\|Xt\|_2^2.$$

*Proof.* Setting $m = \langle Xt, Xg \rangle / \|Xg\|_2^2$ we get

$$\|X(gm - t)\|_2^2 = m^2 \|Xg\|_2^2 + \|Xt\|_2^2 - 2m\langle Xg, Xt \rangle \tag{1}$$

$$= \left(1 - \frac{\langle Xt, Xg \rangle^2}{\|Xt\|_2^2 \|Xg\|_2^2}\right) \|Xt\|_2^2. \tag{2}$$

We use the singular value decomposition of $X = U\Sigma V^T$. Since $g \sim N(0,1)^d$ and $V^T$ is unitary, we have $V^T g \sim N(0,1)^d$ and hence we can assume $V = I$. Then

$$\frac{\langle Xt, Xg \rangle^2}{\|Xt\|_2^2 \|Xg\|_2^2} = \frac{(t^T \Sigma U^T U \Sigma g)^2}{\|U\Sigma t\|_2^2 \|U\Sigma g\|_2^2} \tag{3}$$

$$= \frac{(t^T \Sigma^2 g)^2}{\|\Sigma t\|_2^2 \|\Sigma g\|_2^2} \tag{4}$$

$$= \frac{(\sum_i t_i \sigma_i^2 g_i)^2}{(\sum_i \sigma_i^2 t_i^2)(\sum_i \sigma_i^2 g_i^2)}, \tag{5}$$

where Equation (4) follows from $U^T U = I$ in the SVD. We expand the upper sum to get

$$\mathrm{E}_g\left[\frac{(\sum_i t_i \sigma_i^2 g_i)^2}{(\sum_i \sigma_i^2 t_i^2)(\sum_i \sigma_i^2 g_i^2)}\right] = \mathrm{E}_g\left[\frac{\sum_{i,j} t_i t_j \sigma_i^2 \sigma_j^2 g_i g_j}{(\sum_i \sigma_i^2 t_i^2)(\sum_i \sigma_i^2 g_i^2)}\right] \tag{6}$$

$$= \mathrm{E}_g\left[\frac{\sum_i t_i^2 \sigma_i^4 g_i^2}{(\sum_i \sigma_i^2 t_i^2)(\sum_i \sigma_i^2 g_i^2)}\right]. \tag{7}$$

Here we use the fact that the $g_i$'s are symmetric, so the cross terms of the sum have mean 0. By scaling, we can assume $\sum_i \sigma_i^2 t_i^2 = 1$ and define $p_i = \sigma_i^2 t_i^2$. Then the sum is just a convex combination:

$$(7) = \sum_i p_i \, \mathrm{E}_g \left[ \frac{\sigma_i^2 g_i^2}{\sum_i \sigma_i^2 g_i^2} \right]. \tag{8}$$

Since $\sigma_i \geq \sigma_d$ and $g_i$'s are IID, by direct comparison we have

$$\mathrm{E}_g \left[ \frac{\sigma_i^2 g_i^2}{\sum_i \sigma_i^2 g_i^2} \right] \geq \mathrm{E}_g \left[ \frac{\sigma_d^2 g_d^2}{\sum_i \sigma_i^2 g_i^2} \right]$$

Hence

$$(7) \geq \mathrm{E}_g \left[ \frac{\sigma_d^2 g_d^2}{\sum_i \sigma_i^2 g_i^2} \right] \sum_i p_i = \mathrm{E}_g \left[ \frac{\sigma_d^2 g_d^2}{\sum_i \sigma_i^2 g_i^2} \right].$$

It remains to bound

$$\mathrm{E}_g \left[ \frac{\sigma_d^2 g_d^2}{\sum_i \sigma_i^2 g_i^2} \right] \geq \frac{\sigma_d^2}{\sum_i \sigma_i^2} = \rho, \tag{9}$$

but this follows from a cute, but rather technical lemma, which we will postpone to the end of this section. (Theorem B.4.) $\qquad\square$

It is interesting to notice how the improvement we make each step (that is $1 - \rho$) could be increased to $1 - 1/d$ by picking $G$ from a distribution other than IID normal.

If $X = U\Sigma V^T$, we can also take $g = V\Sigma^{-1} g'$, where $g' \sim N(0,1)^{d_1 \times (k - d_2)}$. In that case we get

$$\mathrm{E} \left( \frac{\langle Xt, Xg \rangle}{\|Xg\|_2 \|Xt\|_2^2} \right)^2 = \mathrm{E} \left( \frac{t^T V \Sigma^2 V^T g}{\|Ug'\|_2 \|Xt\|_2^2} \right)^2 = \mathrm{E} \left( \frac{t^T V \Sigma g'}{\|g'\|_2 \|Xt\|_2^2} \right)^2 = \frac{1}{d_1} \frac{\|t^T V \Sigma\|_2^2}{\|Xt\|_2^2} = \frac{1}{d_1}.$$

So this way we recreate the ideal bound from Theorem B.1. Note that $\frac{\|X\|_{-2}^2}{\|X\|_F^2} \leq 1/d_1$. Of course it comes with the negative side of having to compute the SVD of $X$. But since this is just a theoretical algorithm, it's still interesting and shows how we would ideally update $T_i$. See Figure 6 for the effect of this change experimentally.

It's an interesting problem how it might inspire a better CCE algorithm. Somehow we'd have to get information about the the SVD of $X$ into our sparse super-space approximations.

We now show how to extend the vector case to general matrices.

**Lemma B.3.** *Let $X \in \mathbb{R}^{n \times d_1}$ be a matrix with singular values $\sigma_1 \geq \cdots \geq \sigma_{d_1} \geq 0$. Define $\rho = \sigma_{d_1}^2 / \sum_i \sigma_i^2$, then for any $T \in \mathbb{R}^{n \times d_2}$,*

$$\mathrm{E}_{G \sim N(0,1)^{d_1 \times k}} \left[ \inf_{M \in \mathbb{R}^{k \times d_2}} \|X(GM - T)\|_F^2 \right] \leq (1 - \rho)^k \|XT\|_F^2.$$

*Proof.* The case $k = 1, d_2 = 1$ is proven above in Theorem B.2.

**Case $k = 1$:** We first consider the case where $k = 1$, but $d_2$ can be any positive integer (at most $k$). Let $T = [t_1 | t_2 | \ldots | t_{d_2}]$ be the columns of $T$ and $M = [m_1 | m_2 | \ldots | m_{d_2}]$ be the columns of $M$. Then the $i$th column of $X(GM - T)$ is $X(Gm_i - t_i)$, and since the squared Frobenius norm of a

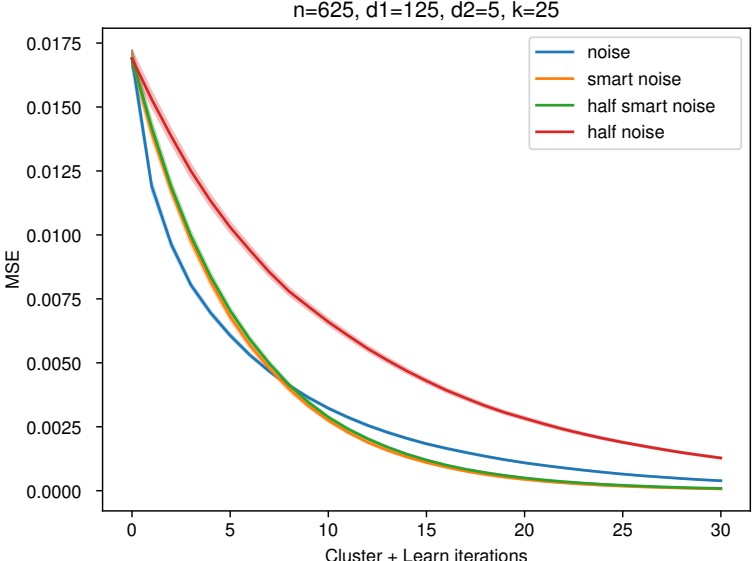

Figure 6: **SVD aligned noise converges faster.** In the discussion we mention that picking the random noise in $H_i$ as $g = V\Sigma^{-1}g'$, where $g' \sim N(0,1)^{d_1 \times (k-d_2)}$, can improve the convergence rate from $(1-\rho)^{ik}$ to $(1-1/d)^{ik}$, which is always better. In this graph we experimentally compare this approach (labeled "smart noise") against the IID gaussian noise (labeled "noise"), and find that the smart noise indeed converges faster – at least once we get close to zero noise. The graph is over 40 repetitions where $X$ is a random rank-10 matrix plus some low magnitude noise.

We also investigate how much we lose in the theorem by only considering $M$ on the form $[I|M']$, rather than a general $M$ that could take advantage of last rounds $T_i$. The plots labeled "half noise" and "half smart noise" are limited in this way, while the two others are not. We observe that the effect of this is much larger in the "non-smart" case, which indicates that the optimal noise distribution we found might accidentally be tailored to our analysis.

matrix is simply the sum of the squared column l2 norms, we have

$$
\begin{aligned}
\mathrm{E}[\|X(GM - T)\|_F^2] &= \mathrm{E}\left[\sum_{i=1}^{d_2} \|X(Gm_i - t_i)\|_2^2\right] \\
&= \sum_{i=1}^{d_2} \mathrm{E}\left[\|X(Gm_i - t_i)\|_2^2\right] \\
&\leq \sum_{i=1}^{d_2} (1-\rho)\,\mathrm{E}[\|Xt_i\|_2^2] \\
&= (1-\rho)\,\mathrm{E}\left[\sum_{i=1}^{d_2} \|Xt_i\|_2^2\right] \\
&= (1-\rho)\,\mathrm{E}[\|XT\|_F^2].
\end{aligned}
\tag{10}
$$

where in (10) we applied the single vector case.

**Case $k > 1$:** This time, let $g_1, g_2, \ldots, g_k$ be the columns of $G$ and let $m_1^T, m_2^T, \ldots, m_k^T$ be the *rows* of $M$.

We prove the lemma by induction over $k$. We already proved the base-case $k = 1$, so all we need is the induction step. We use the expansion of the matrix product $GM$ as a sum of outer products

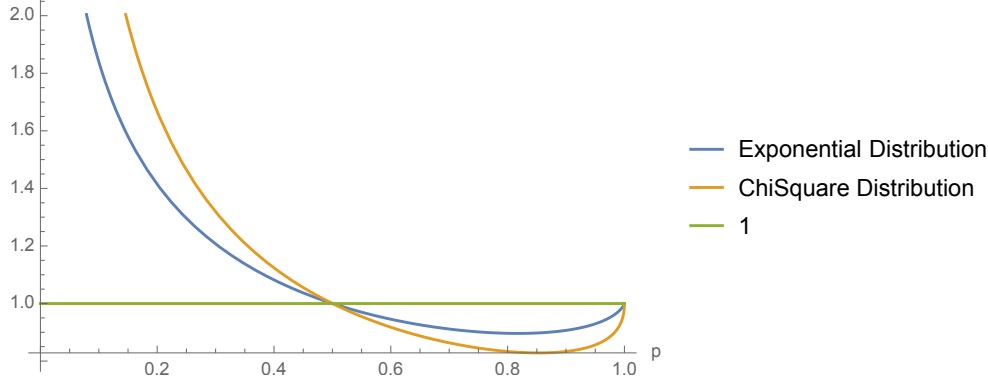

Figure 7: Expectation, $\mathrm{E}\left[\frac{x}{px+(1-p)y}\right]$ when $x, y$ are IID with Exponential (blue) or Chi Square distribution (Orange). In both cases the expectation is $\geq 1$ when $p \leq 1/2$, just as Theorem B.4 predicts.

$GM = \sum_{i=1}^{k} g_i m_i^T$:

$$
\begin{aligned}
\mathrm{E}[\|X(GM-T)\|_F^2] &= \mathrm{E}\left[\left\|X\left(\sum_{i=1}^{k} g_i m_i^T - T\right)\right\|_F^2\right] \\
&= \mathrm{E}\left[\left\|X\left(g_1 m_1^T + \left(\sum_{i=2}^{k} g_i m_i^T - T\right)\right)\right\|_F^2\right] \\
&\leq (1-\rho)\,\mathrm{E}\left[X\left(\left\|\sum_{i=2}^{k} g_i m_i^T - T\right\|_F^2\right)\right] \qquad (11) \\
&\leq (1-\rho)^k\,\mathrm{E}\left[\|XT\|_F^2\right].
\end{aligned}
$$

where (11) used the $k = 1$ case shown above, and (12) used the inductive hypothesis. This completes the proof for general $k$ and $d_2$ that we needed for the full theorem. $\qquad \square$

## B.1   Technical lemmas

It remains to show an interesting lemma used for proving the vector case in Theorem B.2.

**Lemma B.4.** *Let $a_1 \ldots, a_n \geq 0$ be IID random variables and assume some values $p_i \geq 0$ st. $\sum_i p_i = 1$ and $p_n \leq 1/n$. Then*

$$
E\left[\frac{a_n}{\sum_i p_i a_i}\right] \geq 1.
$$

This completes the original proof with $p_i = \frac{\sigma_i^2}{\sum_j \sigma_j^2}$ and $a_i = g_i^2$.

*Proof.* Since the $a_i$ are IID, it doesn't matter if we permute them. In particular, if $\pi$ is a random permutation of $\{1, \ldots, n-1\}$,

$$E\left[\frac{a_n}{\sum_i p_i a_i}\right] = E_a\left[E_\pi\left[\frac{a_n}{p_n a_n + \sum_i p_i a_{\pi_i}}\right]\right] \tag{12}$$

$$\geq E_a\left[\frac{a_n}{E_\pi\left[p_n a_n + \sum_{i<n} p_i a_{\pi_i}\right]}\right] \tag{13}$$

$$= E_a\left[\frac{a_n}{p_n a_n + \sum_{i<n} p_i\left(\frac{1}{n-1}\sum_{j<n} a_j\right)}\right] \tag{14}$$

$$= E_a\left[\frac{a_n}{p_n a_n + (1-p_n)\sum_{i<n}\frac{a_i}{n-1}}\right], \tag{15}$$

where Equation (13) uses Jensen's inequality on the convex function $1/x$.

Now define $a = \sum_{i=1}^n a_i$. By permuting $a_n$ with the other variables, we get:

$$E_a\left[\frac{a_n}{p_n a_n + (1-p_n)\sum_{i<n}\frac{a_i}{n-1}}\right] = E_a\left[\frac{a_n}{p_n a_n + \frac{1-p_n}{n-1}(a-a_n)}\right] \tag{16}$$

$$= E_a\left[\frac{1}{n}\sum_{i=1}^n \frac{a_i}{p_n a_i + \frac{1-p_n}{n-1}(a-a_i)}\right] \tag{17}$$

$$= E_a\left[\frac{1}{n}\sum_{i=1}^n \frac{a_i/a}{\frac{1-p_n}{n-1}-(\frac{1-p_n}{n-1}-p_n)a_i/a}\right] \tag{18}$$

$$= E_a\left[\frac{1}{n}\sum_{i=1}^n \phi(a_i/a)\right], \tag{19}$$

where

$$\phi(q_i) = \frac{q_i}{\frac{1-p_n}{n-1}-(\frac{1-p_n}{n-1}-p_n)q_i}$$

is convex whenever $\frac{1-p_n}{n-1}/(\frac{1-p_n}{n-1}-p_n) = \frac{1-p}{1-np} > 1$, which is true when $0 \leq p_n < 1/n$. That means we can use Jensen's again:

$$\frac{1}{n}\sum_{i=1}^n \phi(a_i/a) \geq \phi\left(\frac{1}{n}\sum_i \frac{a_i}{a}\right) = \phi\left(\frac{1}{n}\right) = 1,$$

which is what we wanted to show. $\qquad\square$

## C  Correspondence between theory and practice

Theorem 3.1 quite tightly matches the empirical behavior of Algorithm 1 (Dense CCE for Least Squares). In Figure 8, we show the convergence when solving the least squares problem $\mathrm{argmin}_T \|XT - Y\|_F^2$ using the method.

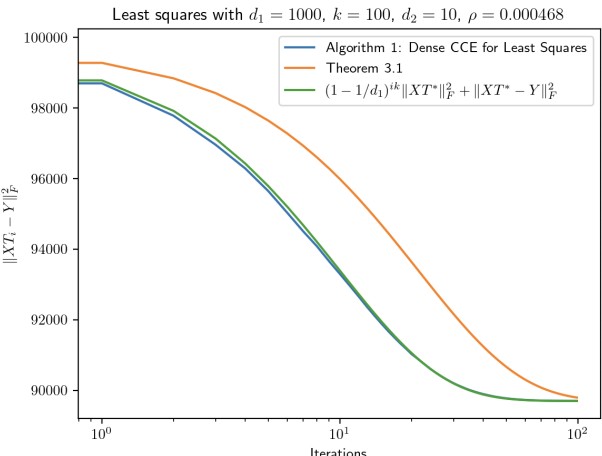

Figure 8: **Convergence of the least squares problem**. We show and compare the convergence of the least square methods using different methods. We found that the convergence of Theorem 3.1 is close to that of Algorithm 1.

As discussed in Appendix B, we can often strengthen the result from using $\rho$ (smallest singular value / the squared Frobenius norm) to just using $1/d_1$. This is because the smallest singular value issue only becomes relevant when Y is chosen to exactly align with the worst possible subspace of X. In practice this is rarely the case. In the particular case of Figure 8, we chose $X$ and $Y$ as iid standard normals.

# D   Hashing

If $h : [n] \to [m]$ and $s : [n] \to \{-1, 1\}$ are random functions, a Count Sketch is a matrix $H \in \{0, -1, 1\}^{m \times n}$ where $H_{i,j} = s(i)$ if $h(i) = j$ and 0 otherwise. Charikar et al. [2002] showed that if $m$ is large enough, the matrix $H$ is a dimensionality reduction in the sense that the norm $\|x\|_2$ of any vector in $\mathbb{R}^n$ is approximately preserved, $\|Hx\|_2 \approx \|x\|_2.$[6]

This gives a simple theoretical way to think about the algorithms above: The learned matrix $T' = H^T T$ is simply a lower dimensional approximation to the real table that we wanted to learn. While the theoretical result requires the random "sign function" $s$ for the approximation to be unbiased, in practice this appears to not be necessary when directly learning $T'$. Maybe because the vectors can simply be slightly shifted to debias the result.

There are many strong theoretical results on the properties of Count Sketches. For example, Woodruff [2014] showed that they are so called "subspace embeddings" which means the dimensionality reduction is "robust" and doesn't have blind spots that SGD may accidentally walk into. However, the most practical result is that one only needs $h$ to be a "universal hash function" ala Carter and Wegman [1977], which can be as simple and fast as the "multiply shift" hash function by Dietzfelbinger et al. [1997].

If Count Sketch shows that hashing each $i \in [n]$ to a single row in $[m]$, we may wonder why methods like Hash Embeddings use multiple hash functions (or DHE uses more than a thousand.) The answer can be seen in the theoretical analysis of the "Johnson Lindenstrauss" transformation and in particular the "Sparse Johnson Lindenstrauss" as analyzed by Cohen et al. [2018]. The analysis shows that if the data being hashed is not very uniform, it is indeed better to use more than one hash function (more than 1 non-zero per column in $H$.) The exact amount depends on characteristics in the data distribution, but one can always get away with a sparsity of $\epsilon$ when looking for a $1 + \epsilon$ dimensionality reduction. Hence we speculate that DHE could in general replace the 1024 hash functions with something more like Hash Embeddings with an MLP on top. Another interesting part of the Cohen et al. [2018] analysis is that one should ideally split $[m]$ in segments, and have one hash function into each segment. This matches the implementations we based our work on below.

---

[6]This also implies that inner products are approximately preserved by the dimensionality reduction.

# E   How to store the hash functions

We note that unlike the random hash functions used in Step 1, the index pointer functions obtained from clustering takes space *linear in the amount of training data* or at least in the ID universe size. At first this may seem like a major downside of our method, and while it isn't different from the index tables needed after Product Quantization, it definitely is something extra not needed by purely sketching based methods.

We give three reasons why storing this table is not an issue in practice:

1. The index pointer functions can be stored on the CPU rather than the GPU, since they are used as the first step of the model before the training/inference data has been moved from the CPU to the GPU. Furthermore the index lookup is typically done faster on CPUs, since it doesn't involve any dense matrix operations.

2. The index pointers can replace the original IDs. Unless we are working in a purely streaming setting, the training data has to be stored somewhere. If IDs are 64 bit integers, replacing them with four 16-bit index pointers is net neutral.

3. Some hashing and pruning can be used as a prepossessing step, reducing the universe size of the IDs and thus the size of the index table needed.

# F   Different strategies for CCE

We include other graphs about CCE in Figure 9. They are all on the Kaggle dataset and were run three times. These graphs helped us develop insights on CCE and choose the correct versions for Figure 4a and Figure 4b.

# G   AUC Graphs

We also evaluate the models using AUC, which is another commonly used metric for gauging the effectiveness of a recommendation model. For example, it was used in [Kang et al., 2021]. AUC provides the probability of getting a correct prediction when evaluating a test sample from a balanced dataset. Therefore, a better model is implied by a larger AUC. In this section, we plot the graphs again using AUC; see Figure 10 and Figure 11.

# H   Table Collapse

Table collapsing was a problem we encountered for the circular clustering method as described in Appendix A. We describe the problem and the metric we used to detect it here, since we think they may be of interest to the community.

Suppose we are doing $k$-means clustering on a table of 3 partitions in order to obtain 3 index pointer functions $h_j^c$. These functions can be thought as a table, where the $(i, j)$-entry is given by $h_j^c(i)$.

There are multiple failure modes we have to be aware of. The first one is column-wise collapse:

| 1 | 0 | 0 |
|---|---|---|
| 1 | 1 | 2 |
| 1 | 0 | 3 |
| $\vdots$ | $\vdots$ | $\vdots$ |
| 1 | 3 | 1 |

In this table the first column has collapsed to just one cluster. Because of the way $k$-means clustering works, this exact case of complete collapse isn't actually possible, but we might get arbitrarily low entropy as measured by $H_1$, which we define as follows: For each column $j$, its column entropy is defined to be the entropy of the probability distribution $p_j : h_j^c([n]) \to [0, 1]$ defined by

$$p_j(x) = \frac{\#\{i : h_j^c(i) = x\}}{n}.$$

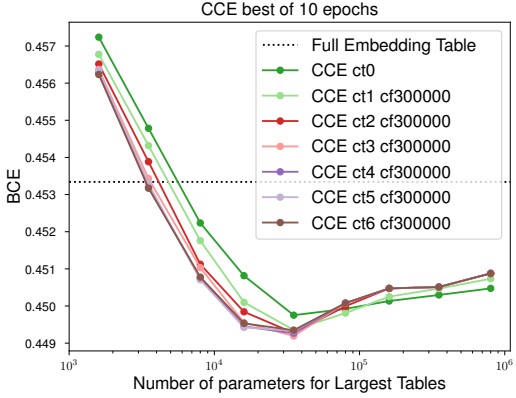

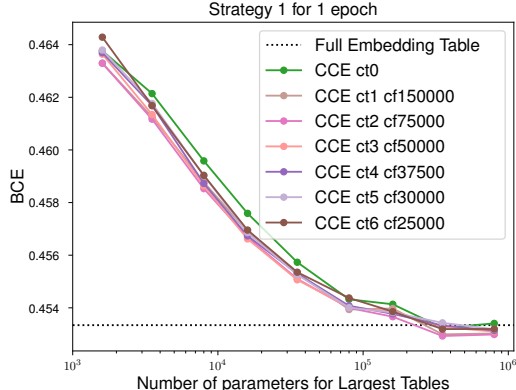

(a) **Kaggle dataset, CCE, best of 10 epoch**: We ran different versions of CCE for 10 epochs. Here ct means the number of clustering done, and cf refers to the number of batches between clusterings. Since each epoch has around $300,000$ batches, we essentially clustered once every epoch. The performance increases with more clustering. Another observation is that as $m$ increases, a few lines were merged due to early stopping. We found that CCE ct6 cf300000 performs the best, which becomes the CCE model in Figure 4a.

(b) **Kaggle dataset, CCE, Strategy 1**: We ran different versions of CCE for 1 epoch under the constraint that all clusterings must finish before half of an epoch. It turns out that there is a balance between the number of clusterings and the 'quality' of the clustering, represented by the number of batches between clusterings. We found that CCE ct2 cf75000 performs the best, which becomes the CCE model in Figure 4b.

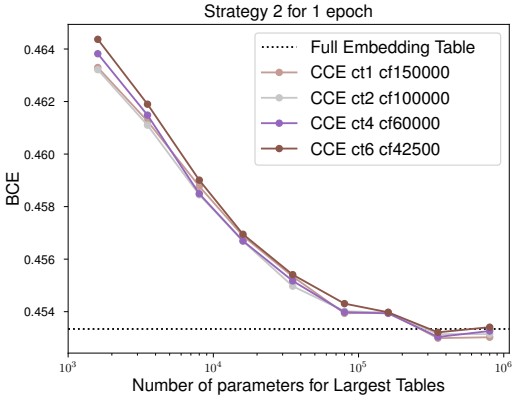

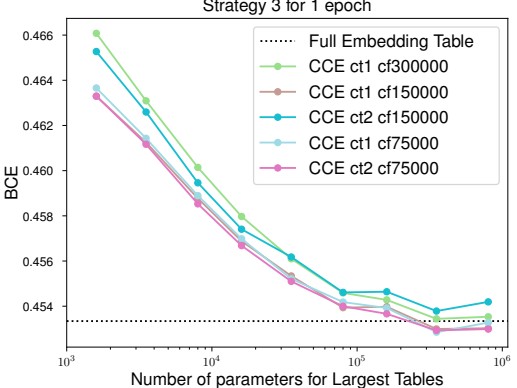

(c) **Kaggle dataset, CCE, Strategy 2**: Strategy 2 here tries to have all the clustering finish by $2/3$ of an epoch. These runs did not perform well. It turned out that we need to let the model have time to converge after all the clusterings.

(d) **Kaggle dataset, CCE, Strategy 3**: This strategy perfectly summarizes all the previous findings. Increasing the number of clusterings in general gives better performance; Letting the model 'rest' after clustering increases the performance; Increasing the interval between clusterings give better result.

Figure 9: Strategies for CCE that gave us insight.

Then we define $H_1$ to be the minimum entropy of the (here 3) column-entropies.

The second failure mode is pairwise collapse:

| | | |
|---|---|---|
| 1 | 0 | 1 |
| 2 | 2 | 3 |
| 1 | 0 | 3 |
| 3 | 1 | 0 |
| 2 | 2 | 1 |

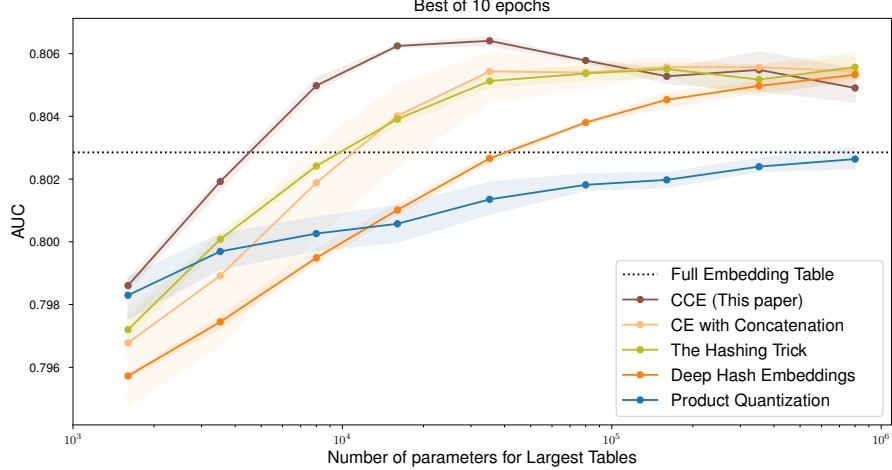

(a) **Kaggle dataset, Best of 10 epoch, AUC**.

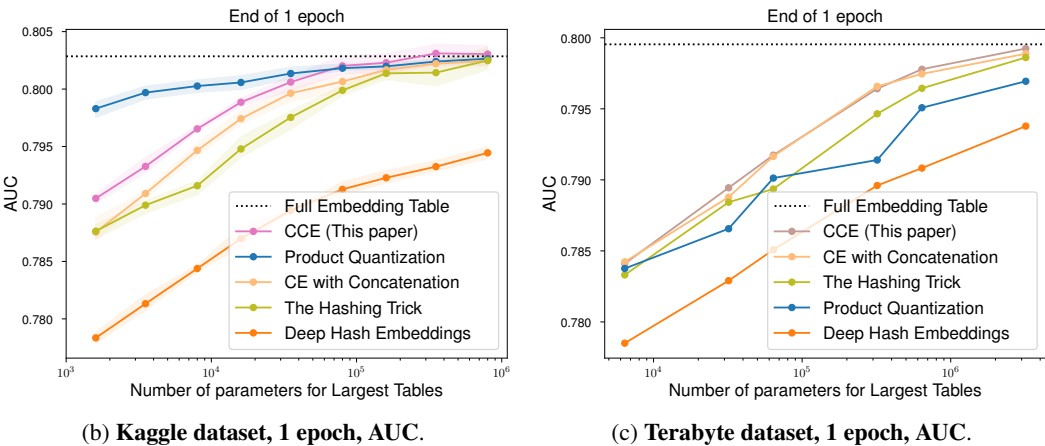

(b) **Kaggle dataset, 1 epoch, AUC**.

(c) **Terabyte dataset, 1 epoch, AUC**.

Figure 10: The AUC version of Figure 4.

In this case the second column is just a permutation of the first column. This means the expanded set of possible vectors is much smaller than we would expect. We can measure pairwise collapse by computing the entropy of the histogram of pairs, where the entropy of the column pair $(j_1, j_2)$ is defined by the column entropy of $h_{j_1}^c(\cdot) + \max(h_{j_1}^c) h_{j_2}^c(\cdot)$. Then we define $H_2$ to be the minimum of such pair-entropies for all $\binom{3}{2}$ pairs of columns.

Pairwise entropy can be trivially generalized to triple-wise and so on. If we have $c$ columns we may compute each of $H_1, \ldots, H_c$. In practice $H_1$ and $H_2$ may contain all the information we need.

### H.1 What entropies are expected?

The maximum value for $H_1$ is $\log k$, in the case of a uniform distribution over clusters. The maximum value for $H_2$ is $\log \binom{k}{2} \approx 2 \log k$. (Note $\log n$ is also an upper bound, where $n$ is the number of points in the dataset / rows in the table.)

With the CE method we expect all the entropies to be near their maximum. However, for the Circular Clustering method this is not the case! That would mean we haven't been able to extract any useful cluster information from the data.

Instead we expect entropies close to what one gets from performing Product Quantization (PQ) on a complete dataset. In short:

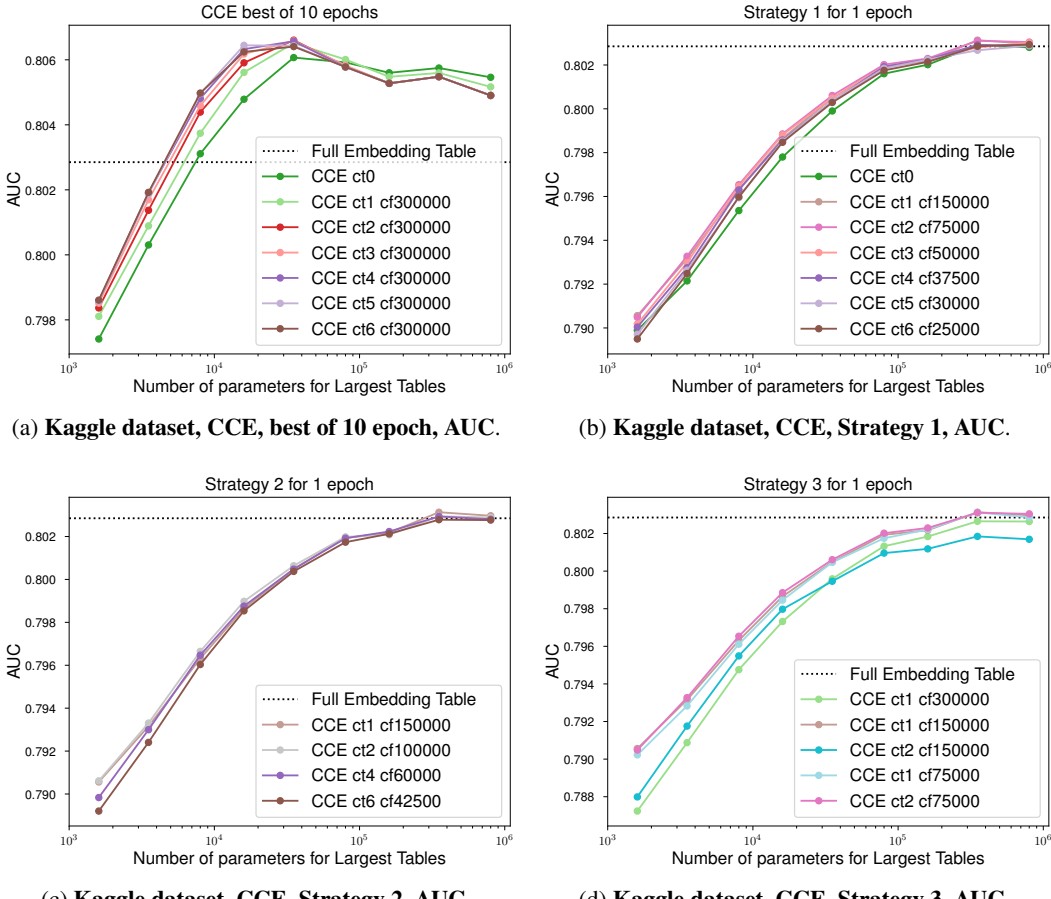

(a) **Kaggle dataset, CCE, best of 10 epoch, AUC**.

(b) **Kaggle dataset, CCE, Strategy 1, AUC**.

(c) **Kaggle dataset, CCE, Strategy 2, AUC**.

(d) **Kaggle dataset, CCE, Strategy 3, AUC**.

Figure 11: The AUC version of Figure 9.

1. Too high entropy: We are just doing CE more slowly.
2. Too low entropy: We have a table collapse.
3. Golden midpoint: Whatever entropy normal PQ gets.

