# OpenReview forum: "Clustering the Sketch: Dynamic Compression for Embedding Tables"
_NeurIPS.cc/2023/Conference — NeurIPS 2023 poster_

### Official Review · Reviewer_jNoe · 2023-07-05

**Soundness:** 3 good
**Presentation:** 3 good
**Contribution:** 3 good
**Rating:** 7
**Confidence:** 3

**Summary:**

In this paper, the authors propose a novel embedding table compression method for recommender models. The novelty of their method, CCE (Clustered Compositional Embeddings) is that it they learn the sparse matrix at training time, instead of using random matrices, as used by other embedding compression methods. It combines clustering and hashing techniques, so as to leverage the high compression rate of the former, and the latter's ability to use the method at training time. CCE achieves baseline accuracy of the full model, with 50% fewer parameters than the SOTA compression models, and 11000 times fewer parameters than the full model. They show experimental results on the click log datasets from Criteo: Kaggle and TB, and show that when trained for multiple epochs, CCE performs at par with the fully learned embedding table, and is always better than other embedding compression methods. The authors also provide theoretical analysis of the method, which shows that for the least squares regression problem, the method is guaranteed to converge to the optimal codebook, and the tight bound on the number of iterations required. The authors also discuss possible extensions of the method, and some interesting insights into how traditional deep recommender models are trained.

**Strengths:**

- Good overview of the problem and related work. The connection of the technique with existing methods, like identifying sparse connections during training, and online product quantization was useful in developing a better intuition of the method.
- The use of good illustrations and pseudo-code improved the readability of the paper.
- Theoretical analysis on the least squares regression problem was sound and strengthened the case for the proposed method.

**Weaknesses:**

- The experimental results only show the Binary Cross Entropy (BCE) loss, however at some places in the paper, the authors say that the model performs baseline accuracy. This is a little confusing. (Upon checking the dataset and its evaluation method, it seems that logarithmic loss is the metric used, so my comment is only about the use of "accuracy" when talking about the results in the paper)


**Questions:**

- Does this technique work well with Language Models? LLMs have huge embedding tables as well, and it seems that this technique could be tried for these models too, leading to good amount of compression.
- What happens when the total number of parameters are the same as the full embedding table? In Fig. 4b, the CCE BCE loss goes below the baseline full embedding table loss. Why is this the case? shouldn't the loss converge at the loss for the full embedding table?
- In Algorithm 3, should it be $M_i'(h_i'(id)$ instead of $M_i'(h_i(id)$? The $h_i'$ is not being used anywhere otherwise.

**Limitations:**

- Experiments on some more datasets would have been useful.
- Discussion on the trade-offs of the method, in terms of any overhead of additional training time or computational resources would be good to see.

---

> ### Author Rebuttal · Authors · 2023-08-08
>
> Thank you for the encouraging feedback!
>
> Re Weakness 1: Thank you for noticing this issue. We do indeed focus on BCE in this paper (and occasionally AUC), but never accuracy. Accuracy is not a very useful measure on our datasets, as they are very biased towards negative samples. We will make sure to fix this confusion in the final paper.
>
> Re Question 1: This is a great question, and there are indeed people who have considered using embedding tables of LLMs, such as the arXiv preprint "TensorGPT: Efficient Compression of the Embedding Layer in LLMs based on the Tensor-Train Decomposition" by Xu et al. However if we consider GPT-2 has 50257 token embeddings of dimension 768, that makes 38.6M parameters. This is only 2.5% of the total 1.5B params. In the paper by Xu et al, they consider GPT-2 “small”, which has only 124M, and so the embeddings become more important, but in general it doesn’t seem to be the bottleneck.
>
> On the flip side, if we have a more efficient way to handle large embedding tables, it might change the tokenization calculus and make it more interesting to use larger tokens / more embeddings than is currently done.
> It is also possible that the regularizing effect of parameter sharing would be useful in avoiding "anomalous tokens" like the famous SolidGoldMagikarp in GPT-3.
>
> Finally, it could be that clustering based methods can help LLMs in other ways, such as giving them cheap access to large tables of information that thus don’t have to be stored in the feed forward layers. However, this is pure speculation and should be investigated in future work.
>
> Re Question 2: In Figure 4(b) the CCE BCE loss goes below the baseline full embedding table loss, even though we only train for 1 epoch. This does indeed look strange, since we don’t expect to overfit one a single epoch training. However, this is probably explained simply by the randomness in independent seeds. We should have added uncertainties to the baseline in addition to the other runs, which we do now in the global response, see Figure 1 of the pdf. Doing this we see that we don’t statistically significantly beat the baseline.
>
> Re Question 3: This is correct. Thank you for catching this.
>
> Re Limitation 1: We ran the open source implementation we’re providing with the paper on Movielens. In particular we implemented a simple DLRM model by embedding the user and the movie, taking the element-wise product and feeding it into an MLP. We then tried to learn whether a user would rate a movie >= 3. We summarize the results in Figure 2 in the globel response pdf, which show the BCE loss for different numbers of table sizes for the 100k and the 1M Movielens datasets.
>
> Re Limitation 2: We discuss some possible limitations of CCE here:
> * Overhead of training time: From our experiments, we did not observe a significant increase in training time: Compared to the hashing trick, CCE saw a 7.5% increase in training time; Compared to CE, CCE saw a 5.5% increase in time.
> * Extra step in the middle of training: CCE requires clusterings, which would pause training until all the clusterings are finished.
> * Need to store the hash functions: Unlike CE, CCE requires storing the index pointer functions. However, we addressed in detail in Appendix D how this is not a problem.

---

### Official Review · Reviewer_mVdt · 2023-07-06

**Soundness:** 4 excellent
**Presentation:** 4 excellent
**Contribution:** 4 excellent
**Rating:** 7
**Confidence:** 4

**Summary:**

This paper discusses the challenges in practical ML when working with large categorical feature tables in modern Recommendation Systems. It introduces a novel approach called Clustered Compositional Embeddings (CCE) that addresses these challenges by combining clustering-based compression techniques with dynamic methods like The Hashing Trick and Compositional Embeddings.

**Strengths:**

1. The proposed CCE offers the benefits of high compression rates from codebook-based quantization while still maintaining the dynamic nature of hashing-based methods. This combination allows CCE to be utilized during training, which is a significant advantage.

2. Theoretical results showing CCE are guaranteed to converge to the optimal codebook. Additionally, it claims to provide a tight bound for the number of iterations required. These theoretical contributions add depth to the proposed method and suggest that CCE is not only practically effective but also supported by solid mathematical foundations.

3. The paper is well-written with organized presentation of CCE.

**Weaknesses:**

NA

**Questions:**

Does the empirical evaluation matches the expectation of theoretical results?

**Limitations:**

The authors adequately addressed the limitations

---

> ### Author Rebuttal · Authors · 2023-08-08
>
> Thank you for the encouraging feedback!
>
> Regarding the question: Theorem 3.1 quite tightly matches the empirical behavior of Algorithm 1 (Dense CCE). We have submitted a plot (see Figure 3 in the pdf in the global response) showing the convergence when solving the least squares problem argmin_T||XT-Y||_F^2 using the method.
>
> As discussed in the appendix, we can often strengthen the result from using rho (smallest singular value / Frobenius norm)^2 to just 1/d_1. This is because the smallest singular value issue only becomes relevant when Y is chosen to exactly align with the worst possible subspace of X. In practice this is rarely the case.
>
> In the particular case of our plot we chose X and Y as iid standard normals.

---

### Official Review · Reviewer_5kPE · 2023-07-07

**Soundness:** 4 excellent
**Presentation:** 3 good
**Contribution:** 4 excellent
**Rating:** 8
**Confidence:** 4

**Summary:**

The embedding of categorical sparse features is a very important component of modern recommendation systems (RecSys). These models replace categorical feature values with a d-dimensional vector embedding that is fed into the model. This problem is not unique to recommendation (it also exists in NLP and other areas), but in RecSys the vocabularies are massive and require advanced compression methods for the embedding table learning to be tractable.

This paper proposes the Clustered Compositional Embeddings (CCE) method, which co-learns a hash projection matrix and a set of embeddings. The hash projections are learned by occasionally performing kmeans clustering on the embeddings (to figure out which embeddings to merge together), and the embeddings are learned by standard methods. The result is an algorithm that learns a smart hash while also learning the embedding values. The paper justifies the algorithm with theory and a lot of experiments, showing better performance for CCE on major recommendation benchmarks.

**Strengths:**

**S1. Good results on a significant problem.** Embeddings are the core driver of performance for some of the largest (up to 12T parameters), most complex, most economically significant (nearly all search, advertisement, and recommendation) applications of machine learning today. Embedding learning has been an active research area for about 20 years and it is very hard to make a significant improvement (as evidenced by all the "approaches that didn't work" in the appendix).

**S2. Great experiments.** This paper implements and compares with (nearly) all of the major baselines over the last 8-10 years.
The ablations and wealth of empirical results are highly informative, not only in understanding the CCE method but also prior methods.

**S3. Interesting ideas.** Clustering normally only works after training the embedding representations - even just a few training iterations after clustering will cause performance to degrade. The common approach is "smart hashing" where we know (apriori) which IDs to merge, but this is hard to do in dynamic production setting. This method seems to get around these challenges.

The theory is also a strong point, as it goes beyond the "dimension reduction" arguments that are common in embedding research and analyzes the embeddings in the context of a supervised learning problem. Though, there is one clarification that would improve this (see W2 below).

**Weaknesses:**

The paper is quite strong. The weaknesses are minor and easily addressed.

**W1. Clarify the implementation.** For example, Algorithm 3 shows a few methods of the CCE table class, but does not show a full learning loop. How often / when should the clustering occur? According to Algorithm 2, it happens many times in a loop, but according to the experiment section we can get away with doing it just once per epoch - or even once per several epochs. I understand that these results are in the appendices but it would be great to summarize the takeaway in the main text. Many of these questions could be also addressed by providing a public implementation alongside the paper.


**W2. Clarify the theory.** The theory applies only to the dense CCE method, which is not actually used in practice. It seems that the dense CCE method might have been developed solely because it is analyzable - this is fine, but it would be nice to see an explicit discussion of why we expect Algorithm 2 to approximate Algorithm 1. If my understanding is correct, this is because we suppose that $T_i$ is low-rank, and use kmeans to approximate the column space of $T_i$ so that the argmin in Line 8 is cheaper (but this point should be explicit). The theory also applies to a regression problem that is not the same as the classification problems used in recommendation, so this limitation should be clarified.

**Questions:**


- How might this work in dynamic settings, where the vocabulary drifts during training / updating? This kind of setting is common in recommendation systems and is one of the reasons why the hashing trick is so widely used (it can easily accommodate more vocabulary entries on the fly).

Other notes:
- Thank you for including your negative results in the appendix. Too few papers do this.
- Figure 5 is referenced in the main text but appears in the appendix. This is likely because a longer form of the paper was cut down to page limit, but it might be good to fix.
- L185: "In line 5 note we only assignments" -> "In line 5, note that we only have assignments"
- L256: Regarding the use of the same embedding table for all features, this idea was recently explored in [[arXiv 2023]](https://arxiv.org/abs/2305.12102) with positive results.

**Limitations:**

Yes, no issues.

---

> ### Author Rebuttal · Authors · 2023-08-08
>
> Thank you for the encouraging remarks. They are truly appreciated.
>
> Re Weakness 1: We have submitted a public implementation of the algorithm, which hopefully can clarify some technical details.
>
> But to clarify the clustering issue: If we think of the leastsq line in Algorithm 2 as completing a full training run to convergence, clustering once per epoch in Algorithm 3 is actually _more often_ than the theory suggests, rather than less. We do this because large scale ML systems are often not trained to convergence even once.
> We have to appeal to an intuition that most of a full training run only finetunes the weights, but we can learn something about the relevant “similarity structure” even from a partially trained model.
> Similarly, most information of the previous epoch intuitively doesn’t need to be thrown away. By seeding the tables with the cluster centroids after clustering, we speed up training in practice.
>
> Perhaps even more surprising is the fact that we can start clustering after _less_ than a full epoch. This is of course necessary for CCE to be relevant in the regime of “single epoch” trained models, like the terabyte Criteo.
> Presumably the reason this works is that even though much of the training data still hasn’t been seen, most of the vocabulary still has. As big as the vocabulary is, the number of data points is even larger.
>
> Re Weakness 2: Our original intuition for why T_i can be approximated by k-means is the success of product-quantization, PQ, and codebook learning methods. If we assume it’s possible to compress a model trained with a full embedding table using kmeans, some of this structure should also be available in the approximate table, T_i, learned using hashing.
>
> You are right that this is akin to T_i being low rank, or even stronger, having a sparse, low-rank basis. However, since we assume k > d2, the sparse basis we find using k-means actually has more columns than T_i. (See Figure 5 in the appendix.) We suppose it makes sense that requiring this level of sparsity may require some more basis vectors.
> It’s curious to notice that this “compression” of T_i into a sparse basis is apparently successful in pertaining the relevant parts of the column space, and keep improving until the optimal sparse column space is found. One might expect this process to converge to some local optimum, but in our experience (some of which Figure 1b shows) this doesn’t happen, and we really find the globally optimal sparse basis.
>
> There is definitely room for more theoretical investigation in Algorithm 2.
> We hope the analysis of Algorithm 1 and the comments above sheds light on why we designed it this way. We will include more of these intuitions into the final paper.
>
> Re Question 1: This is a good question. We have two main solutions to the “dynamic vocabulary problem”.
> 1. You can simply assign the new id randomly in both tables. This matches how every id is assigned at the beginning of training. If we store h_i and h’_i as hashmaps, it’s easy to initialize them specifically for the new vocabulary as it arrives. We can also prune vocab that hasn’t been seen/used for a long time.
> 2. You can use a “pre-hash” function, which maps any incoming id to a manageable space. In fact this is already the case for our DLRM experiments, as it is part of the standard preprocessing which we didn’t try to reverse.
> This hash function will typically map to a space which is in between the size of your table and the size of your vocabulary. E.g., if you have a 10^12 vocabulary, and want a 10^6 table size, you may use an outer hash to reduce the vocabulary to 10^9.
> If you do this, there is no such thing as a “new” id.
>
> In both cases it is important to keep clustering regularly, such as every epoch, so the correct clustering of the new id will eventually be learned.
>
> Re Other notes: Thank you! We will make sure to incorporate these in the final manuscript. We will also cite Coleman et al. for the experiments on Unified Embeddings.

---

> > ### Comment · Reviewer_5kPE · 2023-08-11
> >
> > Thanks for your response! These explanations fully address my questions and I am looking forward to a deep dive into your implementation.

---

### Official Review · Reviewer_Xht4 · 2023-07-10

**Soundness:** 1 poor
**Presentation:** 1 poor
**Contribution:** 1 poor
**Rating:** 1
**Confidence:** 5

**Summary:**

It is obvious that this paper should be rejected due to its poor writing. There are many typos and grammatical errors in the paper. Furthermore, the content is poorly organized.

**Strengths:**

no strengths

**Weaknesses:**

It is obvious that this paper should be rejected due to its poor writing. There are many typos and grammatical errors in the paper. Furthermore, the content is poorly organized.

**Questions:**

It is obvious that this paper should be rejected due to its poor writing. There are many typos and grammatical errors in the paper. Furthermore, the content is poorly organized.

**Limitations:**

yes

---

### Author Rebuttal · Authors · 2023-08-08

Adding a pdf of figures for rebuttal.

---

### Decision · Program_Chairs · 2023-09-21

**Decision:**

Accept (poster)

**Comment:**

The paper present Clustered Compositional Embeddings  to compress embedding tables.  Reviewers were very excited about the idea, experiments and novelty of the papes.  The rebuttal improved the paper.  Reviewers support the acceptance of the paper strongly. One reviewer gave a very strong opinion solely based on writing and the review did not provide any further details, as a result the review was ignored.